# Effect of Sintering Temperature and Iron Addition on Properties and Microstructure of High Speed Steel Based Materials Produced by Spark Plasma Sintering Method

**DOI:** 10.3390/ma15217607

**Published:** 2022-10-29

**Authors:** Marcin Madej, Beata Leszczyńska-Madej, Dariusz Garbiec

**Affiliations:** 1Faculty of Metals Engineering and Industrial Computer Science, AGH-University of Science and Technology, 30 Mickiewicza Ave, 30-059 Krakow, Poland; 2Faculty of Non-Ferrous Metals, AGH-University of Science and Technology, 30 Mickiewicza Ave, 30-059 Krakow, Poland; 3Łukasiewicz Research Network–Poznań Institute of Technology, 6 Ewarysta Estkowskiego St., 61-755 Poznan, Poland

**Keywords:** high speed steel, iron, SPS—spark plasma sintering

## Abstract

Attempts were made to describe the effect of the sintering temperature and pure iron powder addition on the properties of HSS-based materials produced by the spark plasma sintering method (SPS). After sintering, their density, hardness, flexural strength, and tribological properties were determined. The sintered materials were also subjected to microstructural analysis to determine the phenomena occurring at the particle contact boundaries during sintering. On the basis of analysis of the obtained results, it was found that the mechanical properties and microstructure were mainly influenced by the sintering temperature, which was selected in relation to the previously tested steel M3/2, adjusted upwards due to its chemical composition. The use of the temperature of 1050 °C allows materials to be obtained with a density close to the theoretical density (97%), characterized by a high hardness of about 360 HB. The addition of iron slightly reduces the hardness and also increases the flexural strength to 577 MPa. There was no diffusion of the alloying elements from the steel to the iron due to the short time of exposure to the sintering temperature.

## 1. Introduction

Powder metallurgy is an attractive technology from the point of view of the production of tool steels because of its great advantage consisting in obtaining steels with a homogeneous microstructure and controlled carbide precipitates, which provide the steel with excellent isotropic properties. High speed steel produced by powder metallurgy has been adopted for the production of cutting tools, ball production, and structural applications due to its good and universal mechanical properties [1]. This technology is also being developed in terms of powder compacting and sintering methods. In the late 1990s and early 2000s, most of the research focused on sintered HSS with the addition of hard ceramics such as titanium carbide TiC [2,3,4], vanadium carbide VC [5], manganese sulfide MnS [6], niobium carbide NbC [7], aluminum oxide Al_2_O_3_ and titanium nitride TiN [8]. The composition was modified in order to obtain a more wear-resistant material based on HSS. Another goal was also to influence the sintering process by activating it with carbon obtained from the dissolution of carbides, which was the case, for example, in the WC carbide, MnS [8].

PM T15 high speed tool steel is cobalt-bearing super high speed steel. A high volume of hard vanadium carbides provides very high wear resistance. The cobalt content provides excellent resistance to softening at high service temperatures (red or hot hardness). These properties translate into extended retention of the sharp cutting edges on tools produced from PM T15. Owing to powder metal (PM) manufacturing, PM T15 is much easier to grind and exhibits impact toughness that is more than double that of traditional high speed T15 steel [2]. One of the advanced powder metallurgy methods used to produce modern materials or composites is spark plasma sintering (SPS). Recently, numerous publications have appeared in the world literature describing the production of various types of materials by this method. Steels, including high speed steels, do not constitute a very popular group of materials sintered by SPS technology; the only distinguishing group is stainless steels. As an example, the work by Gaël Marnier et al. can be cited [9], which showed that SPS technology can be used to produce 316L steel and can achieve the right microstructure. It has also been proven that in terms of corrosion behavior, samples ofpore-free SPSed 316L steel show an increased passive state in a chloride environment compared to the material produced by traditional methods. Another example is the research of Shashank [10], who determined the sliding behavior without lubrication of duplex and ferritic stainless steels in relation to a diamond counter-sample in a tribological test. The milled nano-structured stainless steel powders were SPS consolidated at 1050 °C. The SPS technique was found to increase the density, hardness and wear resistance of both stainless steels. A. Muthuchamy et al. presented in their publication [11] the impact of alloying additions, such as copper, carbon, and molybdenum with carbonyl iron powder, on the densification behavior, microstructural evolution, and mechanical properties of spark plasma sintered (SPS) compacts. It was shown that after sintering at the temperature of 1120 °C; it was possible to obtain a material with a density of approximately 97%, with good hardness (97 HRB) and a beneficial effect of the molybdenum addition on grain refinement.

The sintering of high speed steel HS 6-7-6-10-0.1LaB6 by the powder metallurgy method (PMHSS) was successfully achieved by the SPS method for the first time, as described by Pellizzari et al. in publication [12]. The influence of the powder particle bonding process in terms of temperature, pressure, and sintering time on the evolution of the microstructure and tensile strength of the samples was investigated in detail. The results indicate that temperature plays a key role in the efficiency of the sintering process. Complete compaction can be achieved at 1100 °C in various pressure and time ranges. When sintering at 1100 °C, under 30 MPa for 30 min, the tensile strength of the sample after heat treatment reaches 2184 MPa, which is more than that of the base material. Additionally, transmission electron microscopy (TEM) investigations of the microstructures of the sintered materials were carried out.

Pellizzari et al. also described the possibility of producing hybrid tool steel with good properties that can be adjusted depending on the specific application [13]. Powder metallurgy (PM) offers the possibility of mixing and sintering different powders to produce a hybrid material that combines the properties of the base materials. Hot work tool steel (HWTS) and high speed steel powder (HSSP) were selected to produce the new steel having high hardness and good toughness. Four mixtures of different compositions (HWTS-HSS: 20–80%, 40–60%, 60–40%, 80–20%) were prepared by SPS and subjected to thermal treatment. The influence of the composition, grain size, and oxygen content was assessed by means of the density, hardness, and apparent fracture toughness of the mixtures. By selecting powders with a small size and a narrow particle size distribution, blends with good mechanical properties can be sintered at almost full density. Large particles make effective co-sintering difficult due to different powder compaction kinetics. The high oxygen content in the base powders does not significantly affect the final density, but has a negative effect on the consolidation process, strongly reducing the strength, especially of HWTS.

The main purpose of this study was to determine the sinterability by means of the SPS method of T15 steel, and T15 steel with an addition of iron, used to reduce the production costs due to its much lower price. The main emphasis was placed on determining the influence of the sintering temperature on the microstructure of the investigated materials. There are no data in the literature on the SPS of this type of steel. Previous studies by the authors on the SPS of M3/2 grade steel proved the applicability of this method; hence, an attempt was made to apply it to another type of tungsten steel.

## 2. Materials and Methods

In the experiments, water-atomized T15 grade powder and Höganäs NC 100.24 iron powder, both finer than 160 µm (more than 78% of the particles are less than 65 µm) were used. HSSP was used in the annealed condition. Its chemical composition is shown in Table 1 and its morphology in Figure 1. Some sellers have introduced T15 powders without molybdenum to the market, some with a small addition of up to 1%. The powder used for the research contains a small addition of it (Table 1).

In high speed steels produced by PM technology, the microstructure of the powder particles is very important; as a result of the low temperatures used in the production process with this technology, it is easy to maintain it in the finished product. Their microstructure consists of fine carbides (MC and M6C type) deployed in a ferritic/bainite matrix. The MC type carbides are visible as finer, brighter particles, while the M6C carbides are larger and gray in color. The typical microhardness of the particles is 292 ± 16 HV0.065. The microstructures of the powders used to produce the study materials are shown in Figure 2.

The powder mixtures of 50 wt% Fe with 50% wt% HSS were prepared by mixing for 60 min in a Turbula T2F (WAB, Muttenz, Switzerland) shaker-mixer and then they were sintered in an HP D 25/3 (FCT Systeme, Effelder-Rauenstein, Germany) furnace. For this purpose, tools made of fine-grade 2334 (Mersen, Gennevilliers, France) graphite were used. The loading chamber in the set of graphite tools was filled with the powder mixture. To enhance the conductivity of the contacts and to prevent sticking, Papyex N998 (Mersen) graphite foil was placed between the powder mixture, the die and the punches. The set of tools prepared in this way was placed in the vacuum chamber of the SPS furnace to carry out the sintering process. 

The powder mixtures were sintered at temperatures of 950, 1000 and 1050 °C and at the compaction pressure of 50 MPa for 2.5 min under vacuum of 5·10^−2^ mbar. The heating rate was 100 °C/min and the pulsed current on:off ratio was set at 125:5 (in ms). Samples with dimensions of Ø40 × 10 mm were produced. 

Subsequently, microhardness (Vickers method, 9.81 N, PN-EN ISO 6507-1:1999), and flexural strength (PN-EN ISO 7438) testing was conducted on the as-sintered samples as well as the density was measured by the Archimedes method (ASTM-D-792). Additionally, the samples were subjected to microstructural examinations by scanning electron microscopy using an SU-70 (Hitachi, Tokyo, Japan) microscope. The phase identification of the composites was carried out by means of a TUR M-62 (Carl Zeiss, Jena, Germany) X-ray diffraction apparatus with Cu radiation (Kα, λ = 1.5406 Å). Three samples were spark plasma sintered under the same conditions, then the properties were determined for each of them, and afterwards the standard deviation was calculated. In the three-point bend test, a rectangular sample was stressed, and the corners were subjected to maximum stresses and strains. Failure occurred when the deformation or elongation exceeded the material limits. The individual stages of production and subsequent investigations are summarized in the diagram below (Figure 3).

## 3. Results

The first step in the production process was to determine the optimal sintering conditions. The optimal compacting pressure was established as 50 MPa, while the temperature was altered in the range of 950–1050 °C in increments of 50 °C. In work [14], the sintering temperatures for the M3/2 steel in the range of 900–1000 °C were estimated; for tungsten steel, this temperature will be higher due to the effect of the activation of molybdenum sintering at the grain boundaries, which is absent in the T15 steel. In the traditional way, a fully dense material made of HSS powders can be obtained by sintering at a temperature close to the solidus line with a sintering window of a maximum of ±5 °C [2]. In most high speed steel grades, the structural components that should be present in the sintered shape after reaching the optimal sintering temperature are as follows [1]: austenite + MC and M6C carbides + liquid phase.

### 3.1. Properties of As-Sintered Materials

The properties of the sintered samples were investigated, which allowed the proper course of their production to be established. The properties of the as-SPSed materials are shown in Figure 4 and Figure 5.

Figure 3 presents the final relative density (measured by the Archimedes method) of the materials as-SPSed. As shown, the T15 and T15-Fe materials with a relative density of 86.4% to 99% (almost a solid material) were obtained by SPS. It is clearly shown that the density is a function of the sintering temperature, iron addition, and the properly selected compacting pressure. Most high speed steels are materials extremely sensitive to the sintering temperature because of the narrow sintering window, which makes it difficult to select the sintering temperature in the SPS method. The complex phenomena at the grain boundary of the powders, resulting from the flow of pulsed current during SPS, give rise to the possibility of the formation of a liquid phase, but without the typical phenomena associated with the sintering of supersolidus. Owing to the duration of the impact of the pulses, the liquid phase probably disappears quickly; hence, its action is short-lived. However, with increasing temperature and simultaneous compaction pressure, full densification becomes possible. Based on the obtained results, it was confirmed that the sintering temperature of 1000 °C is close to the optimal sintering temperature for HSS-Fe materials by the SPS method, as the average relative density is 98%. The factor that hinders the sintering of Höganäs fittings with an addition of iron may be the non-metallic inclusions present in this powder (probably slag) and the internal insulated pores in the iron powder as well. 

This temperature is approximately 250 °C lower than the optimal sintering temperature for high speed steel M3/2, in the range of 1245–1255 °C, which is quite a wide sintering window for high speed steels [2]. This is one reason that could lead to greater interest in SPS technology for HSS-based materials. Nevertheless, the high cost of the process and the low possibility of mass production inhibit the implementation of this technology on an industrial scale.

Table 2 summarizes the results of the microhardness measurements for individual materials depending on the type of microstructure component and the location relative to the grain boundary.

The results of the microhardness measurements presented in Table 2 confirm the absence of diffusion of the alloying elements and carbon from the steel to the iron. The microhardness in the iron particles at the grain boundary of the steel and iron is only slightly higher (about 7 HV0.01), but the difference in hardness is within the limits of statistical error. A similar situation occurs in the areas of high speed steel; the microhardness is similar in the whole grain, and its slight decrease (also within the limits of statistical error) may result from decarburization rather than the dissolution of carbides and diffusion of the alloy elements released in this way. The short pulse effect of temperature is not conducive to diffusion; it can be assumed that the carbides located close to the surface of the high speed steel powder particles may dissolve. Nonetheless, the short sintering time prevents long-range diffusion. Therefore, the effect of increasing the hardness mainly associated with carbon diffusion [15,16], e.g., in composites produced by vacuum infiltration, is not achieved, where the action of elevated temperature is much longer.

Figure 5 presents the results of the flexural strength tests. As in the case with the hardness, the flexural strength also depends mainly on the relative density; with its increase, the flexural strength also grows. However, the HSS-Fe materials sintered at 950 °C and 1000 °C have similar levels of flexural strength. In PM materials, a porosity greater than 5% is a critical value for their mechanical properties. In the studied case, both materials have a relative density above 95% and it is difficult to expect diffusion phenomena during sintering; thus, their flexural strength is comparable. Increasing the sintering temperature to 1050 °C causes a significant increase in the flexural strength, higher in the case of the material with the addition of iron. As the density increases, the importance of the addition of iron also rises since it is plastic, as the main component of the microstructure is ferrite, which has a beneficial effect on the flexural strength. It has been shown that in the case of tungsten steels, including the investigated T15, the iron content is more important than the final density. The difference in the density of these materials is only about 3% (Figure 4).

### 3.2. Microstructures of HSS-Fe Materials

The as-sintered samples were subjected to microstructural examinations by means of both SEM and X-ray diffraction. Typical SEM microstructures of the sintered materials are shown in Figure 6, Figure 7 and Figure 8.

The SEM micrographs presented in Figure 6, Figure 7 and Figure 8 reveal that the microstructure of the spark plasma sintered HSS-grade T15 based materials and T15+50%Fe, contain a steel matrix with finely dispersed carbides, iron particles when added, and small pores (Figure 6, Figure 7 and Figure 8). It can also be seen that during sintering at 950 °C, the pores present in the microstructure of the steel powders (Figure 1a) remained within the steel grains, increasing the sintering temperature to 1050 °C, which eliminates these pores.

The analysis of the microstructures presented in Figure 6, Figure 7 and Figure 8 indicates that the calculated value of density determined by the Archimedes method is correct because the share of pores observed in Figure 6, Figure 7 and Figure 8 corresponds to this. The porosity observed in the micrographs is fairly evenly distributed in the microstructure, unfortunately in the form of elongated and often pointed gaps, unlike molybdenum steel, in which the pores were rounded [14]. The difference in the shape of the pores translates into the results of bending resistance; pointed pores are the nuclei of cracks facilitating the cracking process, which thus significantly lowers the bending strength of the studied materials. Figure 8a presents the uniform distribution of carbides in the HSS matrix near the grain boundary between the HSS particles and the iron particles, after the sintering of the materials. Observations of the boundaries at the interface of the high speed steel and iron particles revealed the presence of a fine dispersion pore network, as shown in Figure 9. This porosity occurs regardless of the sintering temperature used, and it is difficult to unequivocally state that its share decreases with increasing sintering temperature. This phenomenon may be caused by a significant amount of oxygen on the surface of the high speed steel powder produced by the sputtering method. Oxygen resulting from, among others, carbon reductions from the steel matrix are closed by rapidly growing necks, the short sintering time and the subsequent rapid cooling prevent the diffusion of oxygen from the pores beyond the sintered shape. Detailed analysis of the observed boundary also revealed the presence of iron remelting points, which may indicate the presence of a temporary liquid phase during sintering.

To identify the types of carbides present in the T15 high speed steel, point analysis of the chemical composition, element distribution maps, and X-ray phase analysis were performed. Figure 10 presents the uniform distribution of carbides in the as-sintered microstructures of the HSS matrix.

The analyses of the chemical composition of the carbides present in the steel matrix shown in Figure 10 indicate the existence of the main M6C and MC carbides and the presence of ferrite in the matrix after HSS sintering. Most of the Fe is in the matrix and the grey M6C carbides, while W and V are in the grey MC carbides, and tungsten is also the main component of the M6C carbide. The analyses confirm that these carbides do not have a typical stoichiometry; iron also peaks in the MC type carbides, which is typical for sintered high speed steels. Element distribution maps were also made to determine the distribution of elements in the steel matrix.

The maps of the distribution of the basic elements in the steel examined after sintering presented in Figure 11 reveal that carbon, tungsten, and vanadium are found mainly in the carbides. The steel matrix is strongly depleted in carbon, and manganese is dissolved in it. To thoroughly analyze the type of carbides, X-ray phase analysis was performed, the results of which are shown in Figure 12.

The results of the X-ray analysis showed that the M6C type carbides are Fe_3_W_3_C complex compounds, and the MC type carbides are V8C7 type compounds (Figure 12). Comparing these results with the point analysis of the chemical composition of the carbides (Figure 10), it can be added that small additions of molybdenum dissolve in M6C type carbides, and tungsten also dissolve in MC carbides. The curves for both sintering temperatures are almost identical; there are slight differences for the M6C type carbide. This may indicate slight changes in its share in individual sintered compacts, which may, however, result from the dissolution of this carbide at the surface of the powder particles. This may be the result of an excessive sintering temperature as a consequence of the action of electrical discharges. 

A very important aspect of research on spark plasma sintered materials is the determination of the possible diffusion of the alloying elements; in the case of this work, diffusion from the steel to the iron particles. The probable source of the alloying elements is the dissolved carbide of the M6C type, located at the steel-iron interface. To verify this thesis, linear analysis of the distribution of elements at this limit was performed, as illustrated in Figure 13.

The analysis of the presented lines of element decomposition, especially carbon and manganese, indicates a slight diffusion of manganese into iron (from the steel matrix) and no diffusion of carbon into iron. The lack of carbon and the absence of other alloying elements indicate a negligible proportion of carbide dissolution at the boundary and their evaporation, rather than diffusion. Manganese without carbon is not able to significantly improve the properties of iron, and possible subsequent heat treatment may also be of little use.

## 4. Conclusions

Composites having a matrix of T15 high speed steel with a 50% iron addition obtained by SPS at 950, 1000 and 1050 °C have a relative density of up to 97%, where the density grows with increasing the sintering temperature. The hardness and flexural strength depend on the relative density. The T15-Fe composite obtained at the temperature of 1050 °C has the best density-hardness-flexural strength relationship, the relative density of which is 98%, the hardness is 367 HB, and the flexural strength is 1107 MPa. The microstructure of the obtained composites is typical of materials with a high speed steel matrix—it contains characteristic high speed steel grains with very fine precipitates of MC and M6C carbides in a ferritic (or ferritic-bainitic) matrix and iron grains. Characteristic features of the produced materials, which occur independently of the applied sintering temperature, are the very fine pore network on the original external surfaces of the powder particles and the absence of the phenomenon of diffusion of the alloying elements from the steel to the iron. The sintering time of 2.5 min used in our studies is too short to intensify the diffusion process. It is presumed that extending the time would possibly cause diffusion of the alloying elements released from the dissolved M6C type carbides, located close to the grain boundary between the steel and iron.

## Figures and Tables

**Figure 1 materials-15-07607-f001:**
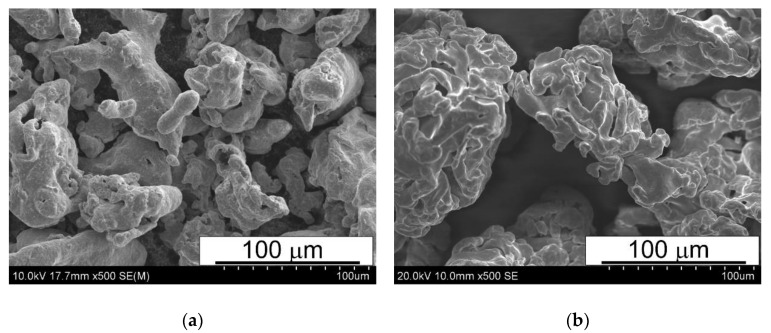
SEM morphologies of powders: (**a**) T15 HSS, (**b**) NC 100.24 iron.

**Figure 2 materials-15-07607-f002:**
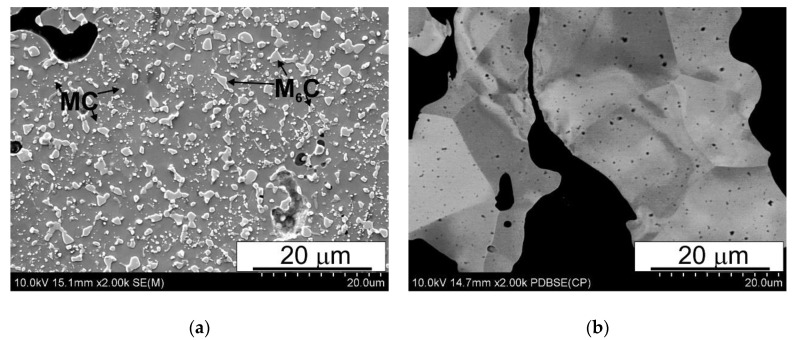
Powder microstructures: (**a**) T15 HSS, (**b**) Fe; SEM.

**Figure 3 materials-15-07607-f003:**
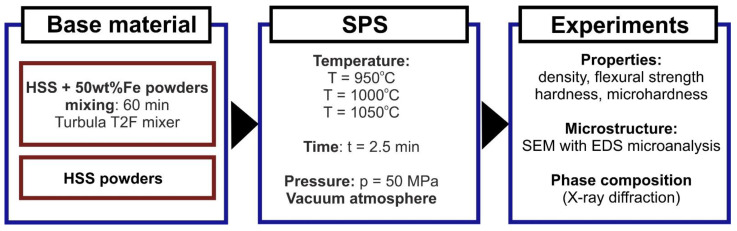
Scheme of T15, T15+50%Fe materials manufacturing process and experimental investigations.

**Figure 4 materials-15-07607-f004:**
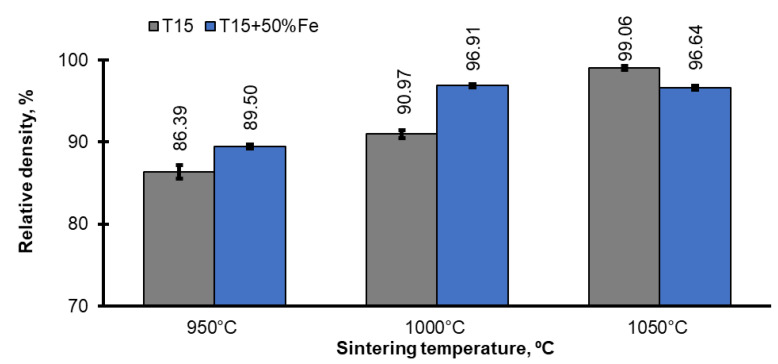
Relative densities of spark plasma sintered T15+50%Fe materials.

**Figure 5 materials-15-07607-f005:**
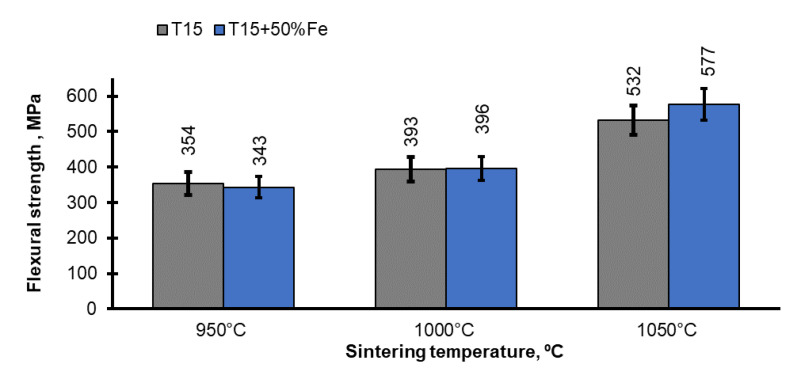
Flexural strength of as-sintered T15 and T15+50%Fe materials.

**Figure 6 materials-15-07607-f006:**
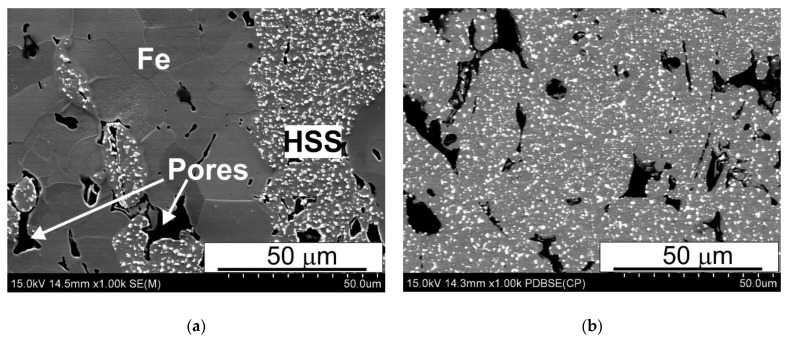
Microstructures of (**a**) T15+50%Fe and (**b**) T15 materials spark plasma sintered at 950 °C; SEM.

**Figure 7 materials-15-07607-f007:**
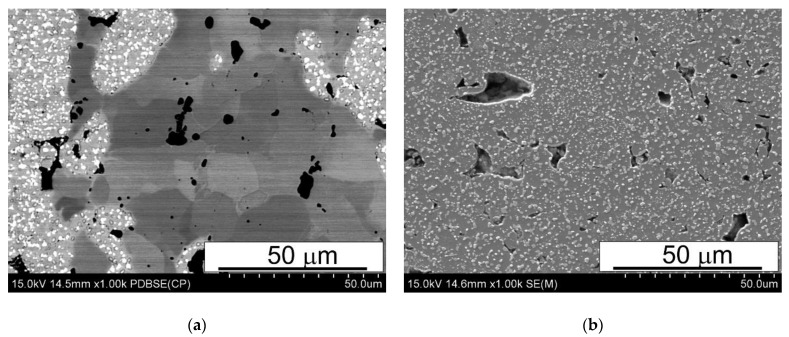
Microstructures of (**a**) T15+50%Fe and (**b**) T15 materials spark plasma sintered at 1000 °C; SEM.

**Figure 8 materials-15-07607-f008:**
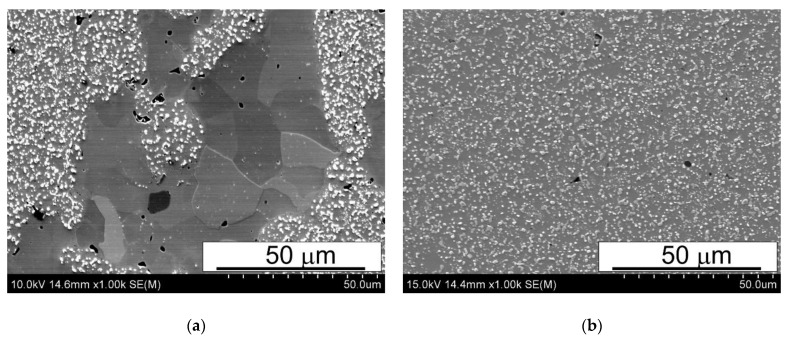
Microstructures of (**a**) T15+50%Fe and (**b**) T15 materials spark plasma sintered at 1050 °C; SEM.

**Figure 9 materials-15-07607-f009:**
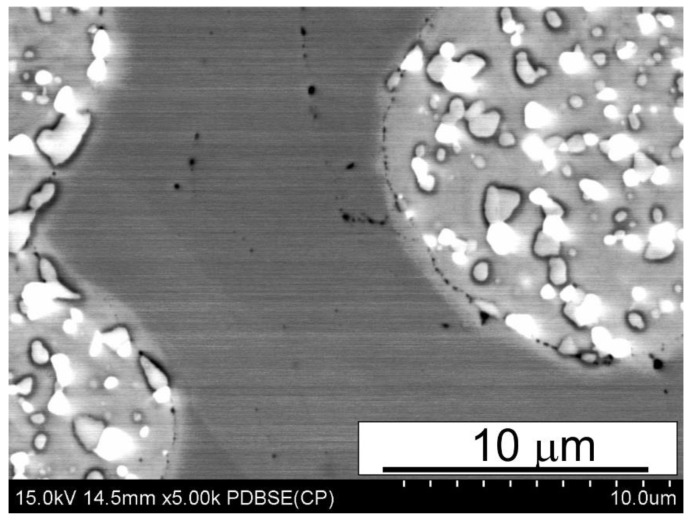
Microstructure of iron-high speed steel grain boundary in T15+50%Fe and b) T15 materials spark plasma sintered at 1050 °C; SEM.

**Figure 10 materials-15-07607-f010:**
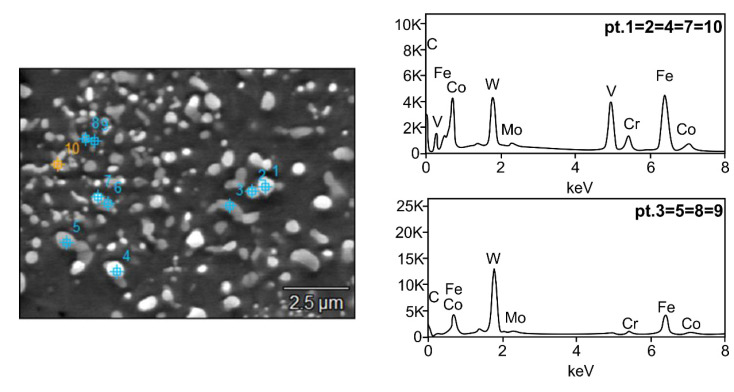
SEM microstructure of HSS matrix of spark plasma sintered T15 steel: 1, 2, 4, 7, 10—MC carbides, 3, 5, 6, 8, 9—M6C carbides.

**Figure 11 materials-15-07607-f011:**
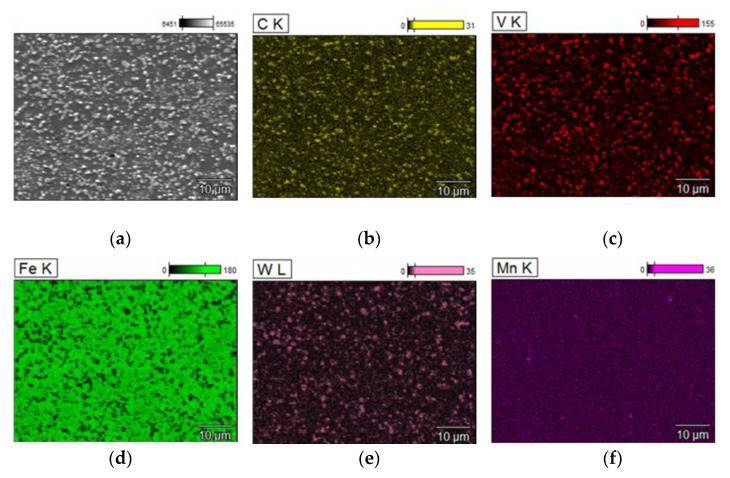
SEM and corresponding EDS mapping micrographs of spark plasma sintered T15 material, (**a**) microstructure, (**b**) carbon, (**c**) vanadium, (**d**) iron, (**e**) tungsten, (**f**) manganese.

**Figure 12 materials-15-07607-f012:**
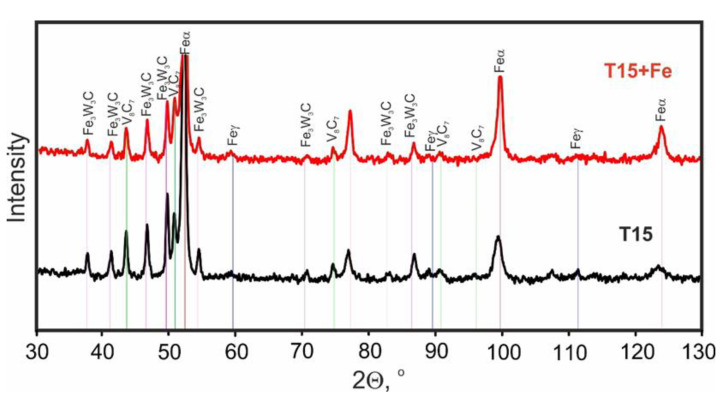
X-ray diffraction pattern of spark plasma sintered T15 and T15+50Fe materials.

**Figure 13 materials-15-07607-f013:**
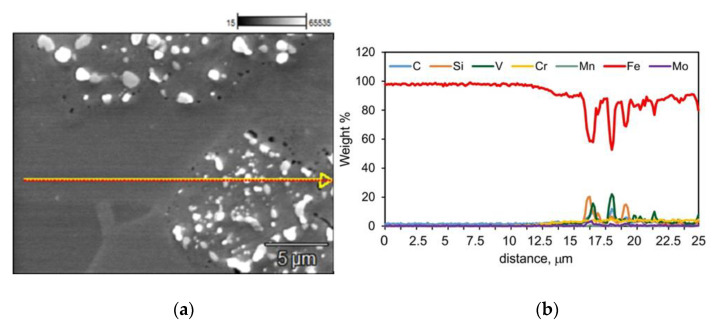
Microstructure of spark plasma sintered T15+50Fe materials, (**a**) microstructure, (**b**) linear analysis of element distribution.

**Table 1 materials-15-07607-t001:** Alloying elements and their proportion by weight in T15 grade steel.

C	Cr	W	Mo	V	Co	Fe
1.6	4	12.25	0.12	4.9	5	rest

**Table 2 materials-15-07607-t002:** Microhardness (HV0.01) of spark plasma sintered T15 and T15+50%Fe materials.

Microstructure Area	Inside Grain	Close Grain Boundary ^1^
T15+50%Fe		
steel	670 ± 19.1	666 ± 17.5
iron	154 ± 17.3	161 ± 16.7
T15		
steel	674 ± 18.5	−

^1^ iron–steel.

## Data Availability

Not applicable.

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
