# Peer review of "Effect of Sintering Temperature and Iron Addition on Properties and Microstructure of High Speed Steel Based Materials Produced by Spark Plasma Sintering Method"

_materials, 2022, doi:10.3390/ma15217607_

Round 1
Reviewer 1 Report
The authors studied the effect of sintering temperature and Fe addition on the properties and microstructure of HSS-based materials produced by SPS method. The aim of this study is to investigate the effect of the sintering temperature and pure iron powder addition on the properties of HSS-based materials produced by the spark plasma sintering method. The manuscript had an interesting topic and was well written, however it could only be accepted with the following minor revisions:
1. Please avoid using acronyms in the paper's title, e.g. Fe, HSS, or SPS.
2. Please see lines 25 and 26, where the phrase looks quite silly (please improve).
3. Line 32 and 33, the authors require to add the abbreviations for TiC, VC, MnS, NbC, Al2O3, TiN, HSS, WC, etc. (please check the whole paragraph and put the abbreviation list before the introduction section is better).
4. Line 67, please add ‘Pellizzari et al.’
5. Line 71, “high speed steel powder (HSS)” could be HSSP
6. For section 2, it is recommended that the authors provide a flow chart for the experimental set up and materials.
7. Figure 10 and 12, please give each image a label.
8. Throughout the paper, it seems more observation works (microstructure). It is highly suggested to do ANOVA analysis for the result of flexural strength and micro hardness. It is useful to identify the significant factor contributing to the experimental conditions. You can refer to these papers (Application of Taguchi Method to Optimize the Parameter of Fused Deposition Modeling (FDM) Using Oil Palm Fiber Reinforced Thermoplastic Composites; Capability of 3D Printing Technology in Producing Molar Teeth Prototype), which can assist you in performing ANOVA analysis.
9. It is suggested to add the limitation, future study, and develop implications for researchers in the conclusion section.
Author Response
Dear Reviewer,
We greatly appreciate your thoughtful remarks that helped improve the manuscript. In the following, we give a point-by-point reply to your remarks. The changes and amendments have been introduced to the text of the publication. All the changes, I have highlight by using the „Track Changes”. The article is also revised by a native speaker.
- Please avoid using acronyms in the paper's title, e.g. Fe, HSS, or SPS.
Reply: Thank you for the comment, the abbreviations have been removed and the full terms have been inserted.
- Please see lines 25 and 26, where the phrase looks quite silly (please improve).
Reply: Unfortunately, part of the sentence disappeared while editing or copying to the form – it has been corrected.
- Line 32 and 33, the authors require to add the abbreviations for TiC, VC, MnS, NbC, Al2O3, TiN, HSS, WC, etc. (please check the whole paragraph and put the abbreviation list before the introduction section is better).
Reply: Thank you for the comment, I introduced full terms of the shortcuts to the text.
- Line 67, please add ‘Pellizzari et al.’
Reply: Thank you for the comment - it has been added.
- Line 71, “high speed steel powder (HSS)” could be HSSP
Reply: Thank you for the comment - it has been changed.
- For section 2, it is recommended that the authors provide a flow chart for the experimental set up and materials.
Reply: Thank you for the suggestion, a flowchart has been inserted into Section 2 (Figure 3).
- Figure 10 and 12, please give each image a label.
Reply: Thank you for your attention. The figures have been corrected, they have new numbers 11 and 13.
- Throughout the paper, it seems more observation works (microstructure). It is highly suggested to do ANOVA analysis for the result of flexural strength and micro hardness. It is useful to identify the significant factor contributing to the experimental conditions. You can refer to these papers (Application of Taguchi Method to Optimize the Parameter of Fused Deposition Modeling (FDM) Using Oil Palm Fiber Reinforced Thermoplastic Composites; Capability of 3D Printing Technology in Producing Molar Teeth Prototype), which can assist you in performing ANOVA analysis.
Reply: Thank you very much for the comment. They are very interesting articles that indirectly explain what ANOVA analysis is. I will try to apply the knowledge in the research that I am currently doing or as part of a doctorate that I am starting with a student on HEA materials produced by powder metallurgy technology.
- It is suggested to add the limitation, future study, and develop implications for researchers in the conclusion section.
Reply: Thank you very much for the suggestion. Being near the optimal sintering temperature, we plan to extend the sintering time to 10 minutes and check the possible diffusion of the steel alloying elements to the iron and the behavior of the carbides - especially the M6C type. Additional text was inserted in lines 430-433.
Reviewer 2 Report
What is the general novelty of the article.There are plenty of works has been carried out in the field of SPS on Ferrous system. The other authors got the apparently good results. ( Refer : https://link.springer.com/article/10.1134/S0031918X18070062
The abstract of the article is written in very general way. The authors did not mention any scientific foundings in the abstract. It should be good practice that the authors must write in details about the results in the abstract. The selection of temperature is important aspect in sintering.
What is the reason to mention the line " Some sellers have introduced T15 powders 92 without molybdenum to the market, some with a small addition of up to 1%. " in the manuscript. Does it carry any significance in the results!
As a reviewer I feel the language of the article must be refined. Some of the sentences are poorly constructed in the manuscript.
on what basis the authors choosen the 50 wt% Fe with 50% wt% HSS in the study? The SPS experimental details were not explained.
The density obtain in the results were not suitable for any structural studies.
The ASTM standard details and size of the specimen for each test must be highlighted in the manuscript.
The test sample details of sample is questionable.
All the figure resolution and the corresponding captions must be improved. Similarly the figure caption also must be improved. The explanation of results must be refined well.
The microstructure does not represent any phase as the authors choose the composition which is not exist for typical application from reveiwers understanding. Also, some of the particles are not sinter well.
Figure 12 is not an XRD plot.
Author Response
Dear Reviewer,
We greatly appreciate your thoughtful remarks that helped improve the manuscript. In the following, we give a point-by-point reply to your remarks. The changes and amendments have been introduced to the text of the publication. All the changes, I have highlight by using the „Track Changes”. The article is also revised by a native speaker.
- What is the general novelty of the article. There are plenty of works has been carried out in the field of SPS on Ferrous system. The other authors got the apparently good results. ( Refer : https://link.springer.com/article/10.1134/S0031918X18070062
Reply: Thank you for the link to the publication on the use of the SPS method for the production of steel-based materials, which I was able to include in the Introduction (line 61-66). I also looked at the publications in the literature in this publication and they will be useful for me in the analysis of research on infiltrated steels, which I also deal with in my research.
- The abstract of the article is written in very general way. The authors did not mention any scientific foundings in the abstract. It should be good practice that the authors must write in details about the results in the abstract. The selection of temperature is important aspect in sintering.
Reply: Thank you for the suggestion. I have completed the abstract regarding the property and briefly indicated the basis for the selection of the sintering temperature. The indicator for the selection of the temperature was the previous research on molybdenum steel M3/2. Knowing the influence of the chemical composition, we found that a higher temperatures should be used and the results for them are presented in this publication. We performed sintering outside our institution, and due to the high cost of the SPS method, we had to narrow the scope of temperature tests.
- What is the reason to mention the line " Some sellers have introduced T15 powders 92 without molybdenum to the market, some with a small addition of up to 1%. " in the manuscript. Does it carry any significance in the results!
Reply: Molybdenum lowers the energy at the grain boundaries, which affects the sintering behavior. Normally, T15 steel was designed so that Mo was not in its composition; however, in analyzing the market, it turns out that such powders are found ever more often.
- As a reviewer I feel the language of the article must be refined. Some of the sentences are poorly constructed in the manuscript.
Reply: The corrected article was sent for review by a qualified native-speaker English language editor.
- On what basis the authors choosen the 50 wt% Fe with 50% wt% HSS in the study? The SPS experimental details were not explained.
Reply: I have been conducting research on high speed steels for about 20 years, which concerns sintering, infiltration and we are currently using the SPS method. I used to add different amounts of iron before, 25, 50 and 75%. The addition of 25% disproportionate to the content reduced the properties of the finished materials. The 75% addition practically eliminated the advantages of high speed steels. On the other hand, the best proportions in costs and properties were obtained for the content of 50% iron. That is why I decided to use such an addition in the presented research. I must admit that the costs of SPS also made it necessary to optimize the parameters and the number of samples used for testing.
- The density obtain in the results were not suitable for any structural studies.
Reply: Unfortunately, this is a constant dilemma when using photos of microstructures of materials characterized by inhomogeneous microstructures, including the distribution of pores. I have tried to include the most characteristic areas that have pores in the viewing area. There are areas where no pores have been found, as well as areas where there is a bit more of them. It is easier in the case of materials with high porosity, and not those where, in relatively accurate measurements using the Archimedes method, we obtain high density. Some sort of justification is also the fact that we estimate the theoretical density of the mixture; therefore, in the case of the samples from the mixture of T15 and iron.
- The ASTM standard details and size of the specimen for each test must be highlighted in the manuscript. The test sample details of sample is questionable.
Reply: The property testing standards have been added to Section 2 (lines 155-157). As for the bending test, we constructed our own holder, which is adapted to samples half the size of the standard, while the rest were performed in a standard way..
- All the figure resolution and the corresponding captions must be improved. Similarly the figure caption also must be improved. The explanation of results must be refined well.
Reply: The figures in the article have been changed according to the additional comments on some of them.
- The microstructure does not represent any phase as the authors choose the composition which is not exist for typical application from reveiwers understanding. Also, some of the particles are not sinter well.
Reply: Thank you for these observations. The composition with the content of 50% T15 - 50% Fe can be treated as more scientific. I dealt with previously infiltrated and classically sintered high speed steels with various iron additives. This content turned out to be the least unfavorable for the final properties in proportion to the content, so I decided to try to sinter this mixture using the SPS method.
- Figure 12 is not an XRD plot.
Reply: Thank you for noticing the incorrect description, it has been corrected.
Reviewer 3 Report
Manuscript Number: Materials 1939557
Title: Effect of sintering temperature and Fe addition on the properties and microstructure of HSS-based materials produced by SPS method
This paper attempts to study the effect of the sintering temperature and pure iron powder addition on the properties of HSS-based materials produced by the spark plasma sintering method (SPS). The work contains some useful results and can be published but after the major revision. The followings are to be clarified.
1. Introduction
1.1 A technical background of the addition of pure iron powder in the sintering process were not detailed given in the introduction.
1.2 Sentence need to be improved. “steels because of its great advantage consisting in obtaining steels with a homogeneous microstructure and controlled carbide precipitates, which provides the steel with excellent isotropic properties” (page1, line 25~27), “Only a small part of these publications deal with the sintering of steels alone or materials with the addition of steel” (page 2. Line 45~46).
2. Materials and Methods
2.1 Remove the figure text of one SEM in Fig.1 “Figure 1. SEM morphologies of powders: a) T15 HSS, b) NC 100.24 iron; SEM” (page 3, line 99).
2.2 Change the all text format of Figure 2, 5, 6, 7, 8 to same as Figure 1 or Figure 9.
2.3 It is confusing the sentence of “The microstructures of the powders used are shown in Figure 2.” Is Figure 2 powder or sintered powder? Please confirm it and change the figure text!
2.4 It is recommended that indication of MC and M6C in the micrograph to easy understand of your manuscript.
3. Results
3.1 One of the important targets of this study is to investigate the effect of addition of pure iron powder in the sintering process. How and why selected only one 50 %-Fe addition in this study?
3.2 Sentences need to be improve “The SEM micrographs presented in Figures 5–7 reveal that the microstructure of spark plasma sintered HSS-grade T15 based materials, T15+50% Fe, contains a steel matrix with finely dispersed carbides and small pores (Figures 5–7).”
3.3 Using magnified SEM micrograph (ex. 3k~5k x) and indicate the finely dispersed carbides of MC and M6C in the micrograph of corresponding Figure 5, 6, 7, 8 and 9.
3.4 Sentence needs to be improve “Figure 8. Microstructure of grain boundary iron-high speed steel in T15+50%Fe and b) T15 materials spark plasma sintered at 1050 @C; SEM.”
3.5 Sentence needs to be improve “Figure 12. XRD pattern of spark plasma sintered T15 and T15+50Fe materials.” It is not XRD pattern and only one materials (may be T15+50Fe)!
4. Conclusions
4.1 There is no specific descriptions for the effect of Fe addition in the conclusions. What we can expect with >50% Fe addition or < 50% Fe addition?
4.2 Use the relative density of 97% instead of 98% for T15-Fe composite in the conclusions.

Author Response
Dear Reviewer,
We greatly appreciate your thoughtful remarks that helped improve the manuscript. In the following, we give a point-by-point reply to your remarks. The changes and amendments have been introduced to the text of the publication. All the changes, I have highlight by using the „Track Changes”. The article is also revised by a native speaker.
Introduction
- A technical background of the addition of pure iron powder in the sintering process were not detailed given in the introduction.
Reply: We have introduced a brief explanation of the motivation for the addition of iron to the investigated steel (lines 99-100).
- Sentence need to be improved. “steels because of its great advantage consisting in obtaining steels with a homogeneous microstructure and controlled carbide precipitates, which provides the steel with excellent isotropic properties” (page1, line 25~27), “Only a small part of these publications deal with the sintering of steels alone or materials with the addition of steel” (page 2. Line 45~46).
Reply: Thank you very much for noticing the errors that arose as a result of editing the text for the template of "Materials". The sentences have been changed. Such a sentence was also inserted (line 56): “Steels, including high speed steels, do not constitute a very popular group of materials sintered with SPS technology, the only distinguishing group being stainless steels.”
- Materials and Methods
2.1 Remove the figure text of one SEM in Fig.1 “Figure 1. SEM morphologies of powders: a) T15 HSS, b) NC 100.24 iron; SEM” (page 3, line 99).
Reply: Thank you for pointing out the repetition in the caption under the figure.
2.2 Change the all text format of Figure 2, 5, 6, 7, 8 to same as Figure 1 or Figure 9.
Reply: The figures have been changed as suggested.
2.3 It is confusing the sentence of “The microstructures of the powders used are shown in Figure 2.” Is Figure 2 powder or sintered powder? Please confirm it and change the figure text!
Reply: It is the microstructure inside the powder particles of both the steel and iron. For steel, it is an area with a large particle size over 180 μm (deliberately selected on a specimen with embedded single powders particles). In line 126-127, the moment of making the powder coating is indicated.
2.4 It is recommended that indication of MC and M6C in the micrograph to easy understand of your manuscript.
Reply: As suggested, the photo has been linked.
- Results
- One of the important targets of this study is to investigate the effect of addition of pure iron powder in the sintering process. How and why selected only one 50 %-Fe addition in this study?
Reply: A brief explanation of the authors' motivation has been inserted, resulting from the aim to reduce the cost of the material itself owing to the difference in the price of the powders.
- Sentences need to be improve “The SEM micrographs presented in Figures 5–7 reveal that the microstructure of spark plasma sintered HSS-grade T15 based materials, T15+50% Fe, contains a steel matrix with finely dispersed carbides and small pores (Figures 5–7).”
Reply: Thank you for the comment, the sentence has been completed.
- Using magnified SEM micrograph (ex. 3k~5k x) and indicate the finely dispersed carbides of MC and M6C in the micrograph of corresponding Figure 5, 6, 7, 8 and 9.
Reply: In the section on the material for testing, I have supplemented a comment about the color of the individual types of carbides in the photos of the microstructure, and links to the photos have also been inserted.
3.4 Sentence needs to be improve “Figure 8. Microstructure of grain boundary iron-high speed steel in T15+50%Fe and b) T15 materials spark plasma sintered at 1050 @C; SEM.”
Reply: The caption has been corrected - thank you for pointing it out.
- Sentence needs to be improve “Figure 12. XRD pattern of spark plasma sintered T15 and T15+50Fe materials.” It is not XRD pattern and only one materials (may be T15+50Fe)!
Reply: Thank you very much for the comment, the caption has been corrected (now Figure 13).
- Conclusions
4.1 There is no specific descriptions for the effect of Fe addition in the conclusions. What we can expect with >50% Fe addition or < 50% Fe addition?
Reply: Only 50% iron was selected for the study because of its cost. The basis for this choice was my earlier works on the infiltration and sintering of high speed steels with various contents of iron addition. It turned out that the 50% additive degrades the properties of high speed steel the least; hence, there was such an additive in the investigations. I will admit that this question arose in the defense of my doctoral dissertation (on infiltration).
4.2 Use the relative density of 97% instead of 98% for T15-Fe composite in the conclusions.
Reply: Thank you very much for your insight. The value has been corrected.
Round 2
Reviewer 2 Report
To all Authors,
I've read the paper, and the article's revision is quite satisfactory. However, I recommend that the authors utilise expert language improvement services.
Reviewer 3 Report
Accept in Present Form